# Interference of *Pseudomonas aeruginosa* Virulence Factors by Different Extracts from *Inula* Species

**DOI:** 10.3390/ph18121824

**Published:** 2025-11-29

**Authors:** Tsvetelina Paunova-Krasteva, Petya D. Dimitrova, Tsvetozara Damyanova, Dayana Borisova, Milena Leseva, Iveta Uzunova, Petya A. Dimitrova, Viktoria Ivanova, Antoaneta Trendafilova, Ralitsa Veleva, Tanya Topouzova-Hristova

**Affiliations:** 1Department of General Microbiology, Stephan Angeloff Institute of Microbiology, Bulgarian Academy of Sciences, Akad. G. Bonchev Street, Bl. 26, 1113 Sofia, Bulgaria; pdimitrova998@gmail.com (P.D.D.); tsvetozaradamianova@gmail.com (T.D.); daqanara@abv.bg (D.B.); 2Department of Immunology, Stephan Angeloff Institute of Microbiology, Bulgarian Academy of Sciences, Akad. G. Bonchev Street, Bl. 26, 1113 Sofia, Bulgaria; mlesseva@microbio.bas.bg (M.L.); ivetauzunova1010@gmail.com (I.U.); petya_dimitrova@web.de (P.A.D.); 3Institute of Organic Chemistry with Centre of Phytochemistry, Bulgarian Academy of Sciences, Acad. G. Bonchev Str., Bl. 9, 1113 Sofia, Bulgaria; viktoria.genova@orgchm.bas.bg (V.I.); antoaneta.trendafilova@orgchm.bas.bg (A.T.); 4Centre of Competence “Sustainable Utilization of Bio-Resources and Waste of Medicinal and Aromatic Plants for Innovative Bioactive Products” (BIORESOURCES BG), 1000 Sofia, Bulgaria; ralitsa_veleva@biofac.uni-sofia.bg (R.V.); topouzova@biofac.uni-sofia.bg (T.T.-H.); 5Faculty of Biology, Sofia University “St. Kliment Ohridski”, 8 Dragan Tsankov Blvd., 1164 Sofia, Bulgaria

**Keywords:** biofilms, quorum sensing, virulence, plant extracts, skin explants, cytotoxicity

## Abstract

**Objectives**: *Pseudomonas aeruginosa* is an opportunistic pathogen of high clinical relevance due to its ability to form biofilms, its inherent virulence regulated by quorum-sensing systems, and its multidrug resistance. In the present study, we evaluated the inhibitory potential of nine extracts from Inula species (chloroform and methanolic fractions, including a sesquiterpene lactone-enriched fraction) against biofilm formation and virulence-associated traits of *P. aeruginosa* PAO1 and three multidrug-resistant clinical isolates, as well as their cytotoxicity, biocompatibility, and ability to affect cytokine and nitric oxide production in infected skin explants. **Methods**: The following methods were applied: fractionation and extraction of plant extracts; cytotoxicity assessment on HFF cells; crystal violet assay for determining antibiofilm activity; fluorescence microscopy for evaluating biofilm viability; electron microscopy for assessing the 3D structure of biofilms and morphological alterations; inhibition assays of pyocyanin pigment, protease activity, bacterial motility, interleukin-17, and nitric oxide production; histological analysis of mouse skin explants. **Results**: Quantitative analyses of antibiofilm activity revealed that five of the tested extracts inhibited biofilm formation by more than 50%. Structural and functional analyses using confocal laser scanning microscopy and scanning electron microscopy demonstrated a substantial reduction in biofilm thickness, exfoliation of biofilm biomass, the presence of isolated bacterial clusters, metabolically inactive cell populations, and morphological abnormalities associated with cell elongation, invaginations, and polar deformations as a consequence of treatment. In addition, the plant extracts strongly affected virulence factors regulated by quorum sensing. The methanolic fractions from *I. britannica* and *I. bifrons* significantly suppressed pyocyanin synthesis. In contrast, the chloroform fractions from *I. helenium* and *I. spiraeifolia* produced the largest inhibition zones in assays for extracellular protease activity. Furthermore, all chloroform extracts suppressed bacterial motility, with the lowest swarming diameter observed for the chloroform and lactone-enriched fractions from *I. britannica*. The chloroform extracts of *I. helenium* and *I. bifrons*, methanolic extracts of *I. britannica*, and chloroform and methanolic extracts of *I. spiraeifolia* showed relatively low toxicity to normal diploid human fibroblasts. Methanolic and chloroform fractions from *I. britannica* disrupted biofilm integrity and reduced IL-17A and nitric oxide production in infected skin explants. **Conclusions**: All these findings indicate a possible synergistic action of the chemical constituents within the fractions on quorum-sensing regulation, biofilm formation, cellular viability, and modulation of host inflammatory responses.

## 1. Introduction

*Pseudomonas aeruginosa* is a multi-resistant pathogen causing infections in various tissues—wounded skin and soft tissues, eye, ear, urinary tract, lung, joints, and bones. The infection occurs after contact with implantation materials such as cardiac pacemakers, heart valves, joint and orthodontic prostheses, catheters, stents, endotracheal tubes, breast implants, contact lenses, or contaminated water. *P. aeruginosa* causes also respiratory infections in immunocompromised individuals and in patients with chronic disease - cystic 79 fibrosis. Severe cases of nosocomial lung infections have been related to antibiotic-resistant causative pathogens of pneumonias and sepsis [1,2].

*P. aeruginosa* forms biofilms, which are less susceptible to surfactants, disinfectants, antibiotics, and biocides. Biofilms represent bacterial communities enclosed in a matrix that adapts the bacteria to the environmental stress by increasing their survival, antibiotic resistance, and drug tolerance and escapes the host defense mechanisms by alteration of host epithelial cell functions and by triggering non-proper immune responses [2,3,4]. During biofilm formation, *P. aeruginosa* cells located in the nutrient-poor regions exhibit reduced metabolic activity and prolonged doubling times [5]. These more or less latent cells, known as persisters, are responsible for the bacterial tolerance to certain antibiotics. *P. aeruginosa* biofilms are often associated with an increased mutation rate, and the spatial proximity of bacterial cells enhances the possibility of horizontal gene transfer, including genes for antibiotic resistance [5]. Furthermore, Gram-negative bacteria such as *P. aeruginosa* are often more resistant due to the complex cell wall structure, which includes outer membrane lipopolysaccharides, which protect against penetration by antimicrobials. Biofilm formation is described as one of the main virulence mechanisms of this bacterial species, responsible for the onset of chronic infections, facilitating bacterial survival within the host [6]. For years, approaches to affect the biofilm formation in *P. aeruginosa* infections were limited to the use of antibiotics. Recently, more studies have shown the potential of plant-derived extracts and molecules to affect biofilm, microbial motility, adhesion, extracellular matrix formation, quorum-sensing (QS) signaling pathways, and dispersion [7]. Substances with proven antibiofilm activity must possess a unique structure that facilitates easy penetration into the cell and influences QS signaling and the occurrence of synergism with other antibacterial agents. Many of these properties are found in certain natural and synthetic compounds [8,9,10]. Plants serve as natural sources of such bioactive products. They are promising producers of metabolites capable of inhibiting and disrupting biofilms, as well as suppressing QS systems. This activity is due to their chemical diversity and high efficacy. In recent years, metabolites isolated from plants have been increasingly recognized as potential inhibitors of bacterial pathogenicity [8,11,12,13]. In most cases, plant-derived compounds act as agonists or antagonists of autoinducers. QS antagonists bind to receptors, thereby disrupting the proper progression of QS cascade processes [8]. Disruption of the quorum-sensing cascade affects the virulence of bacteria and renders them non-pathogenic. The therapeutic properties are mainly attributed to their secondary metabolites, which vary in type and quantity among different plants and possess diverse biological activities [14,15]. Modern research has demonstrated the ability of plant phytochemicals to: inhibit polymeric matrix formation, suppress cell adhesion, limit the synthesis of virulence factors, induce structural deformations in membranes and the cell wall, inhibit protein binding, suppress motility, reduce c-di-GMP (Cyclic diguanylate) synthesis, influence cell permeability, decrease cell surface hydrophobicity, and modulate efflux pump regulation [7,16,17,18,19,20,21,22,23].

The genus *Inula* (Asteraceae) comprises approximately 100 species distributed in Asia, Africa and Europe (mostly in the southern and eastern parts). Some representatives of this genus (*I. helenium* L., *I. racemosa* Hook.f., *I. britannica* L., *I. japonica*, etc.) are used in folk medicine and especially in traditional Chinese medicine, as expectorants, antitussives, anti-tuberculosis, bactericidal and other remedies [24,25,26]. Numerous phytochemical studies of *Inula* species have shown the presence of various secondary metabolites such as sesquiterpene lactones, flavonoids, mono- and dicaffeoyl esters of quinic acid, phenolic compounds, di- and triterpenoids, sterols, etc. [24,25,26,27,28,29]. Extracts from *Inula* species, as well as compounds isolated from them have demonstrated diverse biological activities such as anti-tumor, anti-bacterial, anti-inflammatory, hepatoprotective, antioxidant, anti-fungal, anti-diabetic, etc [24,25,26,27,28,29].

Four representatives of genus *Inula* from the Bulgarian flora have been selected for this study—*Inula spiraeifolia* L., *I. bifrons* (L). L. and the well-known medicinal plants *I. britannica* L. and *I. helenium* L. In our previous study, it has been found that the extracts of these species can reduce biofilms and pigment synthesis in *Chromobacetrium violaceum* [10]. Among them, the extracts from *I. britannica* have shown significant antibiofilm and anti-quorum-sensing effects with over 80% inhibition. These findings led us to continue investigating the effects of *Inula* extracts on biofilm formation and the virulence-related characteristics of other pathogens.

Herein, we propose a biological approach to circumvent antibiotic resistance through the application of a set of nine extracts derived from species of the genus Inula. The aim of our research is to investigate their inhibitory effect on key virulence factors, including biofilm-modulating activity, inhibition of pyocyanin production, proteolytic activity and motility, changes in 3D biofilm architecture (both at the structural level and on single cells within the biofilm) observed via scanning electron microscopy (SEM), and alteration of cell viability within biofilms assessed via confocal laser scanning microscopy (CLSM), evaluation of cytotoxicity and biocompatibility as well as their ability to affect the cytokine and nitric oxide production in infected skin explants.

## 2. Results and Discussion

### 2.1. Chemical Composition of Inula Extracts

The aerial parts of four Inula species—*Inula spiraeifolia* L., *I. britannica* (L.) L., *I. bifrons* L., and *I. helenium* L.—were successively extracted with chloroform and methanol to obtain corresponding chloroform (ISp1, IBr1, IB1, and IH1) and methanol (ISp2, IBr2, IB2, and IH2) extracts. The phytochemical study of *I. helenium* and *I*. *spiraeifolia* led to identification of various terpenoids and phenolic compounds. (Appendix A). The chemical characterization of *I. britannica* and *I. bifrons* extracts has been reported in our previous work [18,30,31,32,33], and the data are summarized in Appendix A. Thus, β-amyrin, 16-hydroxylupeol palmitate, and β-sitosterol were found in the chloroform extracts (ISp1, IBr1, IB1, and IH1) of all studied species; faradiol palmitate and maniladiol palmitate were found in in ISp1, IBr1, and IH1; and taraxasterol, lupeol, and Ψ-taraxasterol and their acetates and palmitates were found in ISp1 and IH1. The samples showed different chemical profiles regarding the presence of sesquiterpene lactones. Britannin, gaillardin, 11,13-dihydro inuchinenolide B, pulchellin C, and ivalin were found in IBr1, isoalantolactone, isotelekin, 4α,5α-epoxy-10,14-dihydro-1-*epi*-inuviscolide, and inuviscolide; 1-*epi*-inuviscolide was found in IB1; and 4α,5α-epoxy-10,14-dihydro-1-*epi*-inuviscolide was found in IH1. In addition, IB1 was found to contain diterpene acids of the *ent*-kaurene type. The methanol extracts (ISp2, IBr2, IB2, and IH2) of all studied species showed a similar profile regarding phenolic acids (chlorogenic acid and 1,5-, 3,5-, 4,5-, and 3,4-dicaffeoylquinic acids) and differed in the content of flavonoids. Thus, quercetin was found in IBr2, IH2, and ISp2; luteolin was found in IBr2 and ISp2; isoquercetin was found in IH2 and ISp2; and luteolin 7-O-glucoside was found in IBr2, kaempferol, apigenin, and rutin; and hyperoside was found in ISp2. Flavonoids were not registered in IB2.

### 2.2. Cytotoxicity on Human HFF Cells

The chloroform plant extracts of *I. helenium* and *I. bifrons* (IH1 and IB1), methanol from *I. britannica* (IBr2), and chloroform and methanol from *I. spiraeifolia* (ISp1 and ISp2) studied by us showed relatively low toxicity to normal diploid human fibroblasts HFF-1 (Figure 1). The extracts of *Inula spiraeifolia*, *I. helenium*, and *I. bifrons*, as well as the methanol extract of *I. britannica*, did not reach an IC50 value in the tested concentration range. The chloroform extract of *I. helenium* preserved cell viability in over 77% of the cells after 24 h of treatment, and within 48 h, adaptation was observed and survival increased to over 95% (Appendix A). In both types of extracts of *I. spiraeifolia* and the chloroform extract of *I. britannica* and *I. bifrons*, the viability of the treated cells was over 80% at all concentrations tested. This effect was maintained even with longer treatment (Appendix A). No concentration-dependent effect was observed in all tested extracts in the concentration range up to 300 µg/mL. Relatively high cytotoxicity, without pronounced concentration dependence, was observed in the total and lactone-enriched extracts of *I. britannica* (Figure 1). For these two extracts, which showed cytotoxic activity, the IC50 value was 50.57 μg/mL for IBr1 and 40 μg/mL for the lactone-enriched IBr1-SL.

The observed low cytotoxicity combined with good antibacterial or antibiofilm activity are desirable qualities for the studied plant metabolites. The low toxicity of plant extracts to human diploid cells is probably due to the antioxidant activity of polyphenols, which has been established by many researchers [34]. The toxicity of extracts containing lactones on actively dividing normal (non-cancerous) cells could be explained by the established interaction of sesquiterpene lactones with the microtubules of eukaryotic cells, thereby blocking cell division. In human cells, this interaction could also affect other processes that depend on the normal functioning of microtubules, including the intracellular movements of cellular organelles [35].

### 2.3. Screening for Biofilm-Inhibitory Activity of Plant Extracts

In a series of experiments, we examined the ability of methanol and chloroform extracts to act as potential biofilm inhibitors in the strain *P. aeruginosa* PAO1 (Figure 2). The presented results are averages from six replicates (wells) across three independent experiments. From the data obtained on the antibiofilm effect of the tested extracts, five of them demonstrated a distinct inhibitory efficiency above 50%, with the highest inhibition values resulting from the inducible action of the chloroform extracts, ranging from 61% to 71%. A comparative evaluation indicates that within the 24 h treatment period, chloroform fractions exhibit a stronger inhibitory effect on biofilm formation compared to methanol extracts (Figure 2). It is known that chloroform plant extracts demonstrate antimicrobial activity, which varies depending on factors such as the bacterial strain and the presence of specific phytochemicals in the extract; however, at present, the data regarding their biofilm-inhibiting activity are highly limited. Biologically active compounds, including polyphenols, alkaloids, flavonoids, and tannins, are often extracted and can interact with the bacterial cell wall, causing damage to underlying membranes, leading to cell lysis and death, which is considered the main reason for the bactericidal effect. Studies show that treatment with chloroform extracts effectively inhibits the growth of Gram-positive and Gram-negative bacteria such as *Staphylococcus aureus* and *Escherichia coli* [36,37].

Our data on PAO1 biofilms show a remarkable inhibitory potential for *I. helenium*, where the chloroform extract (IH1) achieved 71.50 ± 0.107% inhibition, followed by *I. bifrons* (IB1) and *I. britannica* (IBr1). Based on the chemical analysis of the chloroform extracts of these plants (Appendix A), the presence of the sesquiterpene lactones-4,5-epoxy-10α,14αH-1-epi-inuviscolide and ivalin was detected. Our findings regarding these lactones are consistent with those reported by Xie et al. [38], who identified the same compounds in *Carpesium macrocephalum* and demonstrated their involvement in the inhibition of biofilm formation by *Candida albicans*. Sesquiterpene lactones are plant metabolites with numerous proven biological effects [39]. They are sesquiterpenoids containing a γ-lactone ring and exhibit great structural diversity, including eudesmanolides, guaianolides, germacranolides, pseudoguaianolides, and others [40]. There is evidence for their inhibitory role in biofilm formation, suppression of acyl-homoserine lactone production through which they regulate the quorum-sensing systems in *P. aeruginosa*, antimicrobial activity, and inhibition of the pigment violacein [41,42,43]. Their mechanism of action is determined by the α,β-unsaturated γ-lactone moiety. The presence of a α-methylene group or a α,β-unsaturated lactone ring promotes alkylation through Michael-type nucleophilic addition, leading to irreversible binding with other molecules such as enzymes. This mechanism explains some of their biological activities [44].

The strong potential of *I. britannica* extracts to suppress biofilm development was further confirmed in our previous studies using another Gram-negative strain, *Chromobacterium violaceum* [10]. The present analysis also included an enriched sesquiterpene lactone fraction, for which the obtained data indicated a reduction in biofilm formation of nearly 50% (IBr1-SL—47.14% ± 0.0433). Similar findings were reported by Vacheva et al. [45] for a sesquiterpene-containing fraction derived from the medicinal plant *Arnica montana*, which inhibited biofilm formation in uropathogenic clinical isolates of *Escherichia coli*. In contrast, the lowest inhibitory effect on *P. aeruginosa* PAO1 biofilms was observed for the chloroform (ISp1) and methanolic (ISp2) extracts from *I. spiraeifolia*, showing values around or below 10% inhibition (8.13% ± 8.89 and 10.63 ± 7.23). Furthermore, two methanolic extracts from *I. britannica* (IBr2) and *I. helenium* (IH2), which exhibited more than 50% inhibition of biofilm formation, were found to contain quercetin. This compound is well documented to possess antibiofilm activity against *P. aeruginosa*, *C. violaceum*, *Staphylococcus aureus*, and *Staphylococcus epidermidis* [46,47,48,49]. A review of the literature indicates that scientific data on the effect of plant extracts on biofilm formation, particularly for sesquiterpene lactones, remain limited [41,50,51].

### 2.4. Screening on Biofilms from Clinical Isolates

The spread of biofilms is not limited to their occurrence in natural environments but also extends to their presence in food and household industries. The most serious medical concern is their development on various surfaces in healthcare facilities, as well as on different medical devices. In recent years, accumulating evidence has highlighted the biofilm phenotype as a key factor underlying many chronic infections caused by antibiotic-resistant strains. Substances with proven antibiofilm effects must possess a unique structure that facilitates easy cell penetration, influences quorum-sensing signaling, and exhibits synergism with other antibacterial agents. Many of these properties are found in certain plant extracts and metabolites [8,9,13]. This is attributed to their relatively nontoxic nature, biocompatibility, and accessibility [52]. Plant metabolites affect biofilms through several mechanisms, including substrate deprivation, disruption of the cell membrane, interaction with the adhesin complex and cell wall, restriction of nutrient access to bacterial cells, and inhibition of adhesion and growth, among others [19,53].

To evaluate the biofilm-inhibitory capacity of clinical isolates in the presence of plant extracts, we assessed the differences between the effects of the extracts and the control group, expressed as percentages. For this purpose, a series of clinical isolates from an international reference panel were selected [54]. Screening experiments conducted with selected clinical strains with known biofilm-forming capacity [1] revealed a remarkable reduction in biofilm formation when treated with selected plant extracts (Figure 3A–C). The three clinical isolates tested were treated with selected extracts from *I. britannica* and *I. helenium*. Comparatively, the highest percentage of antibiofilm activity was observed for methanol and chloroform extracts of *I. britannica* and *I. helenium* against the clinical isolate *P. aeruginosa* 39016, selected from patients with ocular keratitis. All tested extracts, except for the methanol extract of *I. helenium* (IH2—84.49% ± 1.92), inhibited biofilm formation by over 85% (Figure 3A). Results for the *P. aeruginosa* PAK strain, when compared to the previous isolate, also supported our hypothesis of high biofilm-reducing activity above 75% (Figure 3B). A somewhat lower, but still significant, inhibitory effect was observed for the Mi162 strain, where inhibition exceeded 60%, and for both *I. helenium* extracts IH1 and IH2, inhibition reached approximately 74% (Figure 3C). Considering the data from all three tested strains, and particularly the last one, Mi162, which exhibits multidrug resistance to 12 antibiotics, our results are highly promising, with potential scientific and applied significance for managing surface biofilm infections or biofilm-contaminated surfaces in the food industry and healthcare facilities.

### 2.5. Evaluation of Biofilm Vitality After Treatment with Plant Extracts

During the experimental activities, as part of the structural–functional effects of the plant extracts, their influence on the viability of single cells and/or cells within the biofilm, during biofilm formation, was also investigated. Fluorescence microscopy analysis of *P. aeruginosa* biofilms revealed significant exfoliating activity of the applied substances (Figure 4). In the control sample, the biofilm was multilayered, with a predominant presence of viable cells stained green. In contrast, in the samples treated with extracts, the ability of the *P. aeruginosa* PAO1 strain to form biofilm was significantly reduced. After the addition of the chloroform extract IBr1, a sparse distribution of cells within the population became evident, with a predominance of non-viable cells. Comparing these data with the quantitative results of the antibiofilm screening in *I. bifrons*, we found almost twice as much biofilm inhibition in the chloroform extract than in the methanol extract, which correlated with the viability analysis after treatment with this extract. Fluorescence analysis showed that single island-like formations, composed mainly of dead cells, were observed with the methanol extract IBr2. Biofilm projections for the enriched lactone fraction (IBr1-SL) also demonstrated significant inhibitory activity, where the biofilm profile consisted primarily of metabolically inactive cells and lacked a multilayered structure in the projection. In our previous studies, we demonstrated the inhibitory effect on cell viability within biofilms from Gram-positive and Gram-negative strains using methanol extracts from *I. salicina*. Interestingly, in this Inula species, no biofilm exfoliation was observed in *P. aeruginosa*, and no regions of reduction were detected within the biofilm community; the biofilm was adherent on the glass but non-viable [22]. In contrast, in the present study, compared with the control sample, we observed not only non-viable cells but also multiple regions lacking attached biofilm cells, distributed individually. This indicates an effect of the extracts on biofilm detachment and dispersion. In such cases, it is believed that the synthesis of compounds in the biofilm matrix stops, initiating degradation of the extracellular polymeric substance due to the rupture of covalent bonds between its components [55], which in our case was probably caused by the treatment with the plant extracts.

### 2.6. Effects of Plant Extracts on the 3D Structure of P. aeruginosa PAO1 Biofilms

The surface topography of biofilms treated with the three extracts from *I. britannica* was examined. Observation of the 3D structure of 24 h biofilms in the control sample revealed flat, so-called “confluent” biofilms composed of a clearly defined channel-like structure. The biofilm morphotype exhibited numerous cells forming multilayered structures, homogeneously covering the substrate without visible morphological deformities (Figure 5A,B). Upon addition of IBr1 at high magnification, distinct deformations were observed, manifested as reduced substrate coverage and fused surface relief of single biofilm cells. Single cells displayed elongated, atypical shapes and abnormal invaginations and/or constrictions (indicated by a white arrow, Figure 5C). The biofilm cultivated in the presence of this extract was disintegrated, consisting of individual cells attached to the substrate in sparse clusters (Figure 5C,D). Treatment with IBr2 resulted in visible changes in bacterial cell structure, expressed as altered shape and unusual “sharpening” of the rod shape at one or both poles—indicated by a white star (Figure 5E). Regarding cell distribution within the biofilm, a significant reduction and pronounced loosening were observed, limited to individual cell clusters (Figure 5F). The IBr1-SL fraction also demonstrated exfoliating effects and biofilm reduction, accompanied by changes in single-cell morphology and the surface architecture of treated biofilms. The process of inhibition and disintegration was clearly visible at low magnification, where biofilm formation was reduced to individual cells forming sparse structures, without clear clustering and widespread distribution (Figure 5H). Structural anomalies in single-cell morphology were observed, including longitudinal invaginations along the length of entire cells (white triangle) and atypical invaginations at various cell ends (white arrow) (Figure 5G). Our data on morphological aberrations of the single cells within biofilms summarize the influence of the extracts on biofilm formation dynamics, overall disintegration, and the presence of cellular changes, indicating a certain level of tolerance of the cells to the plant extracts. It has been demonstrated that the exopolysaccharide layer synthesized during biofilm maturation protects against the penetration of antimicrobial substances [56] The morphological changes observed in our study, resulting from the influence of the plant extracts, likely affect exopolysaccharide synthesis during biofilm formation and restrict cell growth within the biofilm, which may also facilitate the infiltration of the applied substances. In this context, it is known that slow-growing cells are more tolerant to antibiotics [57].

### 2.7. Effects of Plant Extracts on Virulence Factors of P. aeruginosa

The effect on one of the virulence factors was assessed by quantifying the levels of the pigment pyocyanin after 24 h of biofilm exposure under the influence of the plant extracts. Pyocyanin is a blue-green pigment that is considered an important secondary metabolite produced by many *P. aeruginosa* strains. Moreover, it is classified as a quorum-sensing signaling molecule and an important virulence factor in a number of infections caused by *P. aeruginosa* [58]. The production of pyocyanin is due to the expression of two phenazine-specific genes, phzM and phzS, and is controlled by the mechanisms of the quorum-sensing system. Our results show that in the presence of three of the methanol extracts (ISp2, IB2, and IBr2), the effect on the suppression of pyocyanin synthesis is increased compared to the chloroform extracts (Figure 6). The highest inhibition on pigment formation was observed by the methanol extracts of *I. britannica* (IBr2) and *I. bifrons* (IB2). In this context, chemical analysis of the methanolic fraction of *I. britannica* (Appendix A) revealed the presence of luteolin and luteolin-7-O-glucoside. Our findings are consistent with those reported by Geng et al. [59], who demonstrated that sub-MIC concentrations of luteolin exert a significant inhibitory effect on pyocyanin production in *P. aeruginosa*. Evidence also suggests that luteolin suppresses the synthesis of the pigment violacein [46]. Similar data confirming the anti-quorum-sensing activity of *I. britannica* fractions were reported in our previous study, where the inhibitory effect of Inula plant extracts was assessed through a reduction in violacein pigment levels in *C. violaceum* [10]. Additionally, the presence of chlorogenic acid in the methanolic extracts (Appendix A) likely contributes to the enhanced inhibitory effect on pigment production, which has also been confirmed by other authors [60,61]. In a comparative sense, we also report efficacy in the lactone-enriched fraction of *I. britannica* (IBr1-SL) (Figure 6). The influence of methanol fractions on the production of pyocyanin has also been proven in another plant species, *Ageratum conyzoides*, which also belongs to the family Asteraceae [62]. Considering that pyocyanin synthesis is also regulated by the quorum-sensing cascade, our results indicate a potential inhibitory effect of the applied Inula extracts on quorum-sensing mechanisms.

As a next step, we analyzed the ability of the extracts to inhibit the production of extracellular proteases. We cultivated *P. aeruginosa* by surface seeding on petri dishes, where the test extracts were dropped into the grooves of 5 mm wells in the agar. After measuring the diameter of the inhibitory zone in the form of a colorless halo around the wells, we reported the results obtained (Table 1). In comparative terms, a more pronounced inhibitory effect on the synthesis of protease enzymes was observed in the chloroform plant extracts, with the widest inhibitory diameter displayed by the extracts from *I. spiraeifolia* (ISp1) and *I. helenium* (IH1). When comparing the extracts with each other, we found a predominantly small diameter of the zones in the methanol extracts. When comparing the influence on two of the virulence factors controlled by quorum sensing, the inhibition of protease enzymes and biofilm formation, we can conclude that there is a synergistic effectiveness of the chloroform extracts from *I. bifrons*, *I. britannica*, and *I. helenium*.

### 2.8. Effects of Plant Extracts on Swarming Motility

Bacterial motility in its various forms supports and even facilitates pathogen–host interactions, including pathogen invasion. For example, swarming motility aids colonization of the urinary tract and is accompanied by biofilm formation on catheters [63]. Moreover, in addition to rapid migration, swarming biofilm cells upregulate their virulence proteins, including hemolysin, urease, and protease [64]. This “social type of behavior” is controlled and promoted by the quorum-sensing system. For the induction of swarming, surface contact is required. In our case, it is possible that in instances where swarming inhibition was observed, the plant extract interfered with adhesion and thereby reduced the stimuli for the differentiation of swarming cells. By comparing the individual samples with the control, we found that the weakest swarming was recorded in the chloroform fraction of *I. britannica* (IBr1), as well as in the lactone-enriched sample (IBr1-SL) (Figure 7A,B). Furthermore, it is noteworthy that stronger swarming suppression was observed in the chloroform extracts compared with the methanol ones. When comparing the analysis of swarming motility, biofilm inhibition assay, and protease activity, we again confirm the higher antivirulent potential of the chloroform extracts compared with the methanol extracts (Figure 2 and Figure 7; Table 1). Reviewing the literature, it appears that to date, data on swarming inhibition by extracts from the genus Inula have been reported by us in an earlier study. There, we evaluated the swarming motility of *C. violaceum*, where we obtained an interesting correlation with the present data. Despite the different strain identities, two of the extracts showed similar inhibition, although at lower levels in our previous findings [10]. Reports on swarming motility can also be found in studies of an ethyl acetate fraction isolated from *A. nilotica*, where inhibition of *S. marcescens* motility decreased in a concentration-dependent manner [65]. In addition, the application of *Lithrea molleoides* extract at 1000 μg/mL resulted in suppression of *P. mirabilis* swarming during the first 8 h [66]. It has also been demonstrated that the essential oil of *Thymus vulgaris* can suppress swarming motility in three different strains: *Pseudomonas savastanoi pv. glycinea* B076, *P. aeruginosa* PAO1, and *P. syringae* [67].

### 2.9. Effect of the Extracts on P. aeruginosa Biofilm Formation at Murine Skin Explants

In order to evaluate the effect of the plant extracts on *P. aeruginosa* biofilm formation, the dorsal skin from healthy Balb/c mice was collected, and the skin explant’s samples were prepared and used to induce bacterial biofilm formation. After 24 h, the biofilm was treated with the extracts and the lactone-enriched fraction, and histology evaluation of the explants was performed after H&E staining (Figure 8). The culturing of skin explants followed by centrifugation for sample collection was related with a weaker cellular loss and changes in the dermis density structure (see control (healthy skin) vs. PBS (cultured in vitro skin) groups in Figure 8). The biofilm was formed after 24 h and was intact after the sample preparation, indicating that it had a stable matrix structure (see the biofilm group in Figure 8). The methanolic extracts from different Inula sp. inhibited *P. aeruginosa* biofilm formation (Figure 8A). In order to quantify the biofilm changes, we performed additional analyses using the Image J software (National Institute of Health, USA) (Figure 8B). The original pictures were initially gated to include the epidermal and dermal regions of the skin explants, and later, the epidermis was precisely outlined, and the pixel density and area were analyzed after channel splitting and greyscale transformation of the blue channel, as the blue staining labelled the bacterial biofilm at the epidermal skin area (Figure 8A, arrow and Figure 8B). The methanol extracts of *Inula* significantly suppressed the mean density of blue staining on the epidermal surface, showing disturbance of the *P. aeruginosa* biofilm (Figure 8C). The milder inhibitory activity on the bacterial biofilms measured by staining density was observed for ISp2 (Figure 8C). The purity of the extracts and the molecular content definitely affected the extracts’ inhibitory effects on the biofilms. While the chloroform extract (IBr1) had strong antibiofilm action, the lactone-enriched fraction (IBr1-SL) was less efficient in decreasing the mean density of the biofilm (Figure 8C). However, the extracts altered the structure of the epidermis, shown by decreased staining area (Figure 8D), and of the dermis, demonstrated by an increased loss in cellularity (Figure 8A, arrow 2). Despite the fact that methanol extract from *I. spiraeifolia* (ISp2) had a weaker antibiofilm effect in comparison to IB2 and *I. helenium* (IH2) (Figure 8A), it also induced less prominent morphological changes in the dermis (Figure 8C) and reduced the blue-stained epidermal area (Figure 8D). The chloroform and methanol extracts from *I. britanica*, but not the lactone-enriched fraction, diminished the epidermal area significantly (Figure 8D). Overall, the data suggested that the extracts might have toxic effects in addition to their antibiofilm action in murine skin explants.

In the same murine skin explant cultures, the production of the cytokine IL-17A (Figure 9A) and nitric oxide (Figure 9B) in the supernatants was determined. The formation of the biofilm of *P. aeruginosa* was related with production of IL-17A and NO, an observation confirming our previous study on skin explants [53]. The *Inula* methanol extracts ISp2, IH2, and IB2 significantly inhibited IL-17 secretion and NO production by the skin explants with *P. aeruginosa* biofilm (Figure 9A,B). These results correspond to the histological data showing suppression of biofilm formation (Figure 8). IL-17 release was dependent on the purity of the extracts because it was affected strongly by the lactone-enriched (IBr-SL) and methanol extracts (IBr2) but not by the chloroform extract from *I. britannica*. The methanolic extract from *I. britannica* was less effective in inhibiting NO than the lactone-enriched fraction. However, the methanol extract and the fraction showed a dose-dependent increase in NO secretion (Appendix A).

*P. aeruginosa* skin infection can directly damage the skin because of secreted virulence factors by the bacteria and the induction of cytotoxicity [68,69] related to a loss of the epidermis and de-keratinization in the human skin model and also to partial damage to the basement membrane [70] due to the disruption of the cell–cell contacts and tight cellular junctions [71]. Indeed, epithelial polarity is an important factor for these effects of *P. aeruginosa* mediated by a quorum-sensing molecule N-3-oxo-dodecanoyl-L-homoserine lactone (3-Oxo-C12-HSL), which altered epithelial integrity in non-polarized cells but not in intact polarized cells [71]. The biofilm formation has been also associated with alterations of the epidermis micro-structure and function because of the formation of the bacterial matrix. The biofilm cells express 130 unique genes vs. 21 genes expressed only in planktonic cells. However, their transcription levels are low, except for the transcripts associated with the type II hxc secretion system. Other genes 494 (8.87%) are downregulated during biofilm formation—like those genes involved in the in denitrification, assimilation of nitrate, and conversion by nitrite reductases to ammonia and the type VI secretion system [72]. The expression of the *nirS* gene, key for the denitrification pathway and of genes nirF and nirM, with functions of the electron transport chain for the denitrification pathway, decreases strongly, and it has been proposed as a marker to distinguish biofilms from planktonic cells. Thus, altered NO in the environment might be important to sustain the *P. aeruginosa* biofilm metabolism and thus, by increasing NO or nitrite reductase NirS in the bacteria, could be used to inhibit *P. aeruginosa* biofilms formation. NO can be excreted in the media by keratinocytes and other cell types in the epidermis and dermis upon infection and/or inflammatory response. We observed that NO is produced by murine explants with *P. aeruginosa* biofilm that likely affect NO metabolism and gene expression in *P. aeruginosa* during biofilm formation. All plant extracts inhibited NO production that corresponds to the decreased area of biofilm formation. Various molecules in the extracts can contribute to inhibited NO secretion, but the metabolite content of the methanol extracts (IBr2) and lactone-enriched fraction from *I. britannica* (IBr1-SL) may have less suppressive effect on produced NO. Consistently, we found the best efficiency of the lactone-enriched fraction of *I. britannica* (IBr1-SL) in antibiofilm formation in vitro and on the suppression of pyocyanin synthesis (pigment formation), showing an altered quorum-sensing cascade. These data indicate that plant extracts from the *I. britannica* can affect biofilm formation, although they can inhibit the secretion of NO by host skin cells.

The murine skin explants with *P. aeruginosa* biofilm produced IL-17A. The cytokine can alter the transcriptional landscape of fibroblasts, keratinocytes, and skin-resident and infiltrating immune cells [73]. It promotes the inflammatory response in infection and regulates tissue homeostasis and integrity. The plant extracts decreased IL-17A secretion, which may affect several antimicrobial molecules in fibroblasts (such as the antimicrobial peptides S100 calcium-binding protein (S100) A12, S100A7A (psoriasin), serpin family B member (SERPINB) 4, SERPINB3, and lipocalin 2 (LCN2); the proinflammatory cytokines IL1A, IL1B, IL8, IL6, and IL23A; and the chemokines CCL20, CXCL2, CXCL3, and CXCL5) and in keratinocytes [73]. The plant extracts inhibited IL-17A production, probably due to the presence of flavonoids. Some of the numerous examples are kaempferol [74], quercetin [75], and luteolin [76]. The plant extracts decreased IL-17A production, which might compromise the inflammatory response against bacterial biofilm. However, we observed that the extracts inhibited biofilm formation despite this possible effect on the host defense mechanisms. Various metabolites in the extracts may have a cumulative action on suppressing biofilm formation but may also increase cellular toxicity and cell loss in skin explants. For example, IB2 showed a strong antibiofilm activity but disrupted the epidermal and dermal structure and strongly inhibited IL-17A and NO production. Thus, we concluded that the antibiofilm action of the extracts might be uncoupled from their effect on NO and IL-17 production by the skin explants. Various qualitative and quantitative contents of metabolites in the extracts can contribute to these effects.

## 3. Materials and Methods

### 3.1. Plant Material

Plant material was collected in full flowering stage from native populations of the species in Bulgaria during 2017–2019 as follows: *I. spiraeifolia* L. from the Struma river valley, *I. britannica* L. from the Sofia region, *I. bifrons* (L.) L. from the Rhodopes Mts., and *I. helenium* L. from Verila Mt. The species were identified in the Herbarium of the Institute of Biodiversity and Ecosystem Research, Bulgarian Academy of Sciences, by Dr. Ina Aneva according to the following specimens SOM 176695, SOM 172474, SOM 176699, and SOM 169052.

### 3.2. Preparation of the Extracts

Air-dried and powdered aerial parts (20 g) of *I. spiraeifolia*, *I. britannica*, *I. bifrons*, and *I. helenium* were consecutively extracted with chloroform (200 mL, 3 times) and methanol (200 mL, 3 times) at room temperature for 24 h each. The corresponding chloroform (ISp1, IBr1, IB1, and IH1) and methanol (ISp2, IBr2, IB2 and IH2) extracts were obtained after filtration and evaporation of the solvents under reduced pressure. In addition, a portion of the chloroform extract of *I. britannica* (IBr1, 300 mg) was dissolved in 2.5 mL of CHCl_3_ and applied on a glass column containing 30 g of Silica gel 60 (230–400 Mesh, Merck, Germany) equilibrated with CHCl_3_. Elution was performed with CHCl_3_/acetone mixtures in ratios of 10:1 and 8:1 (*v*/*v*). The fractions were controlled by TLC (Silica gel 60 F_254_, Merck, Germany) using the same mobile phases. Detection was achieved by spraying TLC plates with conc. H_2_SO_4_ and heating at 100 °C. Fractions containing sesquiterpene lactones were combined and concentrated under reduced pressure to produce a fraction enriched in sesquiterpene lactones (IBr1-SL, 15.8 mg).

The isolation and identification of individual compounds from *I. britannica* (IBr1 and IBr2) and *I. bifrons* (IB1 and IB2) has been already reported in [18,30,31,33].

The fractionation of the chloroform and methanol extracts from *I. helenium* and *I. spiraeifolia* and the identification of the main compounds are described in the Appendix A.

GC-MS analysis of the triterpenoids was carried out with an Agilent 7890B (Agilent, Santa Clara, CA, USA) gas chromatograph equipped with a flame ionization detector (FID) and mass selective detector (MSD) (Agilent 5977A using HP-5MS capillary column (5%-phenyl)-methylpolysiloxane, 30 m × 0.25 mm; 0.25 μm film thickness, Agilent) under the chromatographic conditions described in [32]. The identification of triterpene alcohols and their acetates was performed by GC-MS and comparison with the literature data [32]. The identification of triterpene palmitates was achieved by alkaline hydrolysis and further GC-MS analysis of the triterpene alcohols and fatty acids [32].

^1^H and ^13^C NMR spectra were recorded on a Bruker Avance II+ 600 NMR spectrometer (Bruker, Bremen, Germany) with operating frequency 600 (^1^H) and 150 (^13^C) MHz in CDCl_3_ or CD_3_OD. The structures of 16-hydroxylupeol-*O*-palmitate, maniladiol palmitate, faradiol palmitate 4α,5α-epoxy-10α,14-dihdyro-1-*epi*-inuviscolide, rutin hyperoside, isoquercetin, quercetin, luteolin, kaempferol, and apigenin were confirmed by comparison of their ^1^H NMR data with those reported in the literature [77,78,79,80] and/or with those of the authentic standards.

### 3.3. Cytotoxicity on Human Cells

The cytotoxicity of plant metabolites on normal human diploid skin fibroblasts (HFF-1, ATCC-SCRC-1041) was assessed by crystal violet staining as described in Trendafilova et al. [81]. The methods, based on the oxidation of tetrazolium salts to formazan or resazurin to resorufin, are not suitable for assessing cell viability in the presence of antioxidants such as many plant metabolites, as the results could be affected by changes in oxidative status upon treatment [60]. Cells were grown for 24 h in a 96-well flat-bottomed plate at a starting concentration of 2 × 10^4^ cells per well. The treatment was performed for 24 h with extracts diluted in Dulbecco’s Modified Eagle Medium (DMEM) at concentrations ranging from 0 to 300 µg/mL. The absorbance was measured at 570 nm using an Epoch Microplate Spectrophotometer, BioTek^®^ Instruments Inc., Winooski, VT, USA, with the Gen5TM Data Analysis software, version 1.11.5.

### 3.4. Strains and Culture Conditions

The activities involved in the present study included the strain *P. aeruginosa* PAO1, as well as three clinical isolates from the International Reference panel (Table 2). Strain maintenance was described previously [82].

### 3.5. Biofilm Inhibition Assay

To evaluate the biofilm-inhibitory properties, an 18 h inoculum from the bacterial cultures (the model strain *P. aeruginosa* PAO1 and three clinical isolates (Table 1)) was cultivated in Trypticase soy broth (TSB) (Himedia, Thane, India) at 37 °C for 24 h. For the determination of the biofilm-inhibitory capacity of the substances, Eppendorf tubes were prepared with 900 μL of minimal salt medium M63 with the following composition (0.02 M of KH_2_PO_4_, 0.04 M of K_2_HPO_4_, 0.02 M of (NH_4_)_2_SO_4_, 0.1 mM of MgSO_4_, and 0.04 M of glucose, pH 7.5) and a final concentration of the plant extracts of 250 μg/mL. To each prepared sample, 10 μL of bacterial inoculum from each strain was added. The final mixtures were vortexed to ensure thorough mixing and then dispensed in 150 μL volumes into 96-well, round-bottomed microtiter plates (Corning^®^ Costar^®^, Corning, NY, USA), in six replicates per sample. To prevent drying during biofilm cultivation, sterile distilled water was added to the peripheral wells. The plates were incubated for 24 h at 37 °C under static conditions. The next step involved removing the non-adhered bacteria by washing each well three times with 150 μL of phosphate-buffered saline (PBS). This was followed by staining with 150 μL of 0.1% aqueous crystal violet solution for 15 min at room temperature. The wells were then repeatedly washed with 150 μL of PBS until the blue tint of the dye was removed. The final step was solubilization for 5 min with 150 μL of 70% ethanol added to each well. The absorbance was measured at 570 nm using an ELISA reader (INNO, Incheon, Republic of Korea).

### 3.6. Live/Dead Staining Protocol

Biofilm viability was evaluated by cultivation of biofilms on borosilicate glasses in the presence of selected plant extracts at a final concentration of 250 μg/mL. A fluorescence microscopy study was conducted to assess biofilm viability after treatment with the extracts. For the experimental procedure, the Live/Dead BacLight™ kit (Invitrogen, Carlsbad, CA, USA) was used. This kit contains a combination of DNA markers: Syto9 (staining all viable bacterial cells—green signal) and propidium iodide (PI—red signal), which penetrates only cells with compromised membrane integrity. The Live/Dead BacLight™ dye was prepared according to the manufacturer’s instructions, and the samples were stained for 10 min. The cover glasses with the samples were mounted in Fluoromount medium for fluorescence microscopy. The prepared specimens were examined under a confocal laser scanning microscope (Nikon Eclipse TiU, Nikon, Tokyo, Japan) in epifluorescence mode with excitation wavelengths of 488 nm and 543 nm, using a 60× immersion plan-apochromatic objective and a Nikon DS-Fi1 CCD camera (Nikon, Melville, NY, USA). Image acquisition (at least 20 images selected at random) was performed using the NIS Elements software package (Ver. 4.0), and further processing was carried out with the image analysis software Icy (Ver. GPLv3).

### 3.7. Scanning Electron Microscopy

An inoculum was prepared from an 18 h culture of *P. aeruginosa* PAO1 in the presence of selected plant extracts at a final concentration of 250 μg/mL. The prepared samples were applied onto sterile polystyrene fragments that had been pre-sonicated at a frequency of 40 kHz at 25–30 °C, treated with ethanol, rinsed with dH_2_O, and placed in a 24-well plate. Cultivation was carried out at 37 °C for 24 h. The formed biofilms were washed with 0.1 M of sodium cacodylate buffer (CB), pH 7.2, followed by fixation for 2 h at 4 °C in 4% glutaraldehyde in 0.1 M of CB. After triple washing in CB, post-fixation was performed for 1 h in 1% OsO4 in 0.1 M of CB, again at 4 °C. This step was followed by dehydration in a graded ethanol series, with 15 min incubation in each ethanol solution. The prepared samples were mounted onto scanning electron microscopy holders using silver conductive tape, air-dried at room temperature, and sputter-coated with gold in a vacuum evaporator (Edwards, Irvine, CA, USA). Observations were performed with a Lyra/Tescan scanning electron microscope (TESCAN GROUP a. s., Brno, Czech Republic) at an accelerating voltage of 20 kV, and images were recorded using the dedicated microscope software.

### 3.8. Inhibition of Pyocyanin Synthesis

To assess one of the significant virulence factors, we performed a test for the inhibition of the pigment pyocyanin by the applied plant extracts. *P. aeruginosa* PAO1 was cultivated in 10 mL of TSB medium for 18 h in the presence of the plant extracts. The synthesized pyocyanin was evaluated through extraction with 5 mL of chloroform. After the formation of two phases, 2 mL from the lower phase were pipetted and transferred into a new sterile test tube. Then, 1 mL of 0.2 M HCl was added, followed by vortex mixing to induce phase separation. After the two phases formed, the pink supernatant was pipetted into a 96-well plate, and absorbance was measured at 520 nm using an ELISA reader (INNO, Republic of Korea).

### 3.9. Protease Inhibition Assay

To evaluate the inhibition of protease activity by the plant extracts, we used Calcium Casein Agar plates (Merck KGaA, Darmstadt, Germany). The strain was surface-inoculated onto the plates. Wells with a diameter of 5 mm were made in each Petri dish using a probe. Then, 20 µL of the plant extracts were dispensed into the wells, and cultivation was carried out at 37 °C for 24 h. Inhibition of protease activity was determined by the appearance of a clear halo around each well containing extract. The diameter of inhibition was measured and compared with the control sample.

### 3.10. Swarming Motility Assay

To assess the swarming motility of *P. aeruginosa* in the presence of the tested plant extracts, a medium consisting of 0.6% TSA supplemented with 0.4% (*w*/*v*) sterilized glucose was prepared. As a positive control, Petri dishes without the addition of extracts were prepared to determine normal migration. Sterile plastic agar plates were inoculated at the center with a bacterial culture and the corresponding extract (*w*/*v*). After inoculation, the Petri dishes were incubated for 24 h at 37 °C. Each sample was tested in duplicate, and the diameters of the swarming motility zones were measured (mm) in comparison with the control sample.

### 3.11. Mice

Male or female inbred Balb/c mice were purchased from the Slivnitsa Animal Farm of the Bulgarian Academy of Sciences (Bulgaria) and bred at the Animal Facility of the Institute of Microbiology (Bulgaria). The animals were housed under 12/12 h light/dark cycle with free access to water and food. All experiments were performed under veterinary supervision and according to national legislation and guidelines (SG Issue 87/2006) and Decree 20 on 1 November 2012: License for Animal Housing #352 on 30 January 2012 (#11130005); License for Experimental Procedures #125 on 7 October 2020 issued by The Animal Ethics Committee at the Bulgarian Food Safety Agency and in accordance with the ARRIVE guideline (Animal Research: Reporting of In Vivo Experiments) and the principles of the 3Rs (replacement, reduction, and refinement).

### 3.12. Murine Skin Explants—Cultivation, Biofilm Formation, and Treatment

Healthy mice (23 ± 5 g, *n* = 3 female, and *n* = 3 male) were used for the preparation of the murine skin explants from a dorsal back skin according to our previously protocol [53]. Briefly, the skin was aseptically removed, cut on smaller samples with a size of 0.5 cm^2^, added at the bottom of 24-well plates (TPP, Trasadingen, Switzerland), and cultured as an air–liquid interface explant in pre-warmed 250 µL/well DMEM containing 4500 mg/L glucose, L-glutamine, sodium bicarbonate, sodium pyruvate (Sigma-Aldrich, Munich, Germany) and 10% fetal calf serum (FBS; F9665; heat-inactivated, non-EU origin; Sigma-Aldrich, Munich, Germany). Then, the samples were washed two times with 250 µL of PBS and then with sterile dH_2_O. They were allowed to dry and infected with 25 µL of *P. aeruginosa* suspension containing 1 × 10^5^ CFU/mL. The biofilm was allowed to form for 24 h. The explants were then washed with sterile PBS and loaded with 50 µL of PBS:DMEM 1:1 or plant-derived extracts dissolved in DMEM at concentration of 100 µg/mL. After 24 h, the plate was centrifuged at 250× *g* for 5 min, and the supernatants were aseptically collected, passed through a sterile syringe with 20 µm pore filters (Milipore, Merck, Darmstadt, Germany), aliquoted, and frozen at −20 °C while the skin samples were fixed and used for histology.

### 3.13. Histology of Skin Explants

The standard histological procedures were performed for skin explants, including dehydration, embedding in paraffin, cutting with microtome, and hematoxylin and eosin (H&E) staining. The stained tissue sections were observed on a Leica DM2000 microscope, and pictures were taken with the microscope camera at 20× and 40× magnifications. The photos were analyzed by ImageJ Software (version 1.8.0_172 (64 bit), National Institute of Health, Bethesda, MD, USA).

### 3.14. Nitric Oxide Production by Skin Explants

Nitric oxide (NO) production by skin explants was determined with colorimetric Griess reaction as previously described [53]. Briefly, supernatant samples were thawed at 37 °C, ultrafiltrated, and diluted 1:5 with PBS, and 100 µL of the samples were incubated with the same amount of Griess reagent. The absorbance was measured by Innova microplate reader (Infitek, Irvine, CA, USA) at wavelength of 550 nm. The concentration of NO was determined from a standard curve of NaNO_2_ (Sigma-Aldrich, Munich, Germany), and the concentration of Nitrite (in nM) was calculated for each sample according to the formula described previously [53].

### 3.15. Secretion of IL-17A by Skin Explants

IL-17 production by skin explants and its secretion in the cultures was determined with ELISA kit according to the manufacturer’s protocol (sensitivity range of 16–1000 pg/mL; Peprotech, London, UK) and according to the protocol of Damyanova et al. [53]. The absorbance at 405 nm with a wavelength correction at 650 nm was measured by Innova microplate reader (Infitek, Irvine, CA, USA). The concentration of the cytokine in the samples was calculated from a standard curve of recombinant mouse IL-17A.

### 3.16. Statistical Analysis

To validate the quantitative results for biofilm and pyocyanin inhibition, statistical significance (* *p* < 0.05, ** *p* < 0.01, *** *p* < 0.001; *n* = 6) was determined relative to the control group (bacterial inoculum without addition of plant extracts) using one-way analysis of variance (ANOVA) and marked with asterisks. The same statistical method was used for data analyses of cytotoxicity, infected skin explants, NO, and IL-17A production. Data are presented as mean values ± standard deviation (SD), analyzed with OriginPro 9.0 software followed by Tuckey’s and Dunnett’s tests to statistically evaluate the differences between individual experimental groups.

## 4. Conclusions

The obtained data clearly emphasize the trend that plant extracts from Inula species exhibit significant antibiofilm and antivirulence potential against *P. aeruginosa*. Chloroform fractions demonstrated the strongest inhibitory effects on biofilm formation, extracellular protease synthesis, and motility, whereas methanolic extracts effectively suppressed pyocyanin production. Structural–functional analyses of the 3D biofilm architecture confirmed the presence of cellular deformations and overall biofilm disintegration, as well as reduced viability and sparse cellular distribution. Their low cytotoxicity and the ability to simultaneously disrupt biofilm formation, reduce cell viability, alter biofilm architecture, suppress quorum-sensing-dependent traits, and modulate the host inflammatory responses position these extracts as a promising source of natural compounds for the development of novel antivirulence strategies aimed at limiting biofilm-associated infections and enhancing therapeutic efficacy against multidrug-resistant *P. aeruginosa* strains.

## Figures and Tables

**Figure 1 pharmaceuticals-18-01824-f001:**
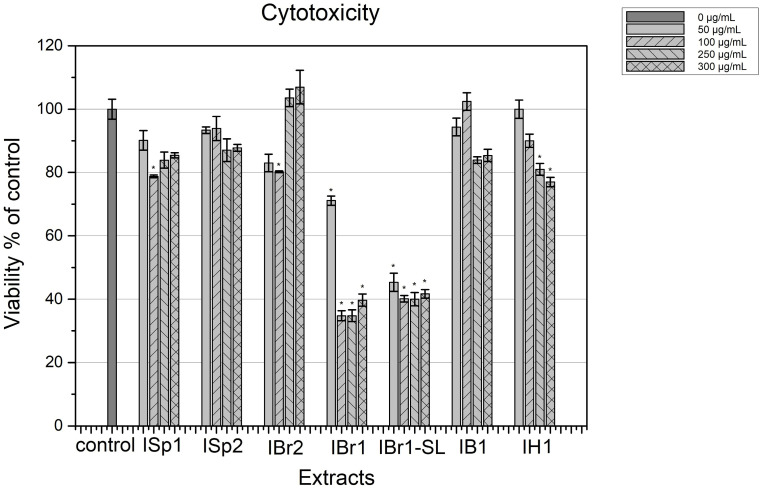
Cytotoxicity of selected plant extracts on HFF cells. Extracts of *I. spiraeifolia*—ISp1 (chloroform) and ISp2 (methanolic); *I. britannica*—IBr1 (chloroform) and IBr2 (methanolic); IBr1-SL (lactone-enriched fraction of IBr1). Chloroform extracts of *I. bifrons*—IB1 and *I. helenium*—IH1; control—untreated cells. * *p* < 0.05—statistically different inhibition of cell viability compared with control group.

**Figure 2 pharmaceuticals-18-01824-f002:**
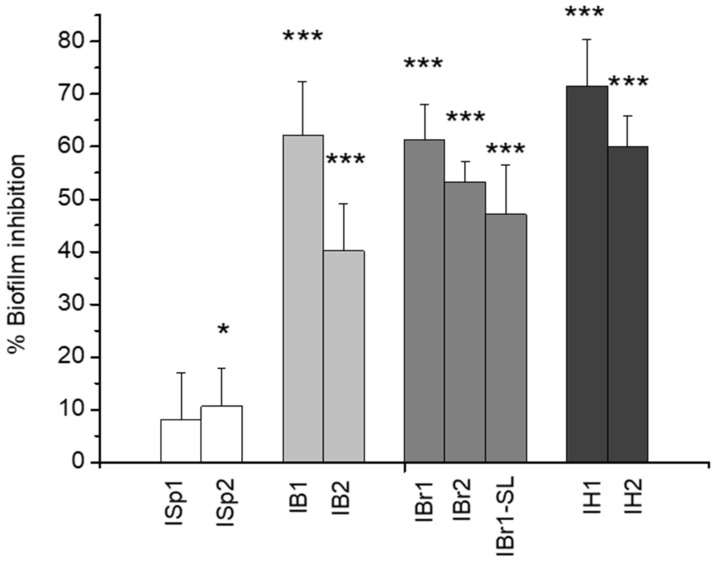
Evaluation of the inhibitory efficacy of plant extracts on *P. aeruginosa* biofilm. Statistically significant effects are indicated with asterisks. * *p* < 0.05; *** *p* < 0.001.

**Figure 3 pharmaceuticals-18-01824-f003:**
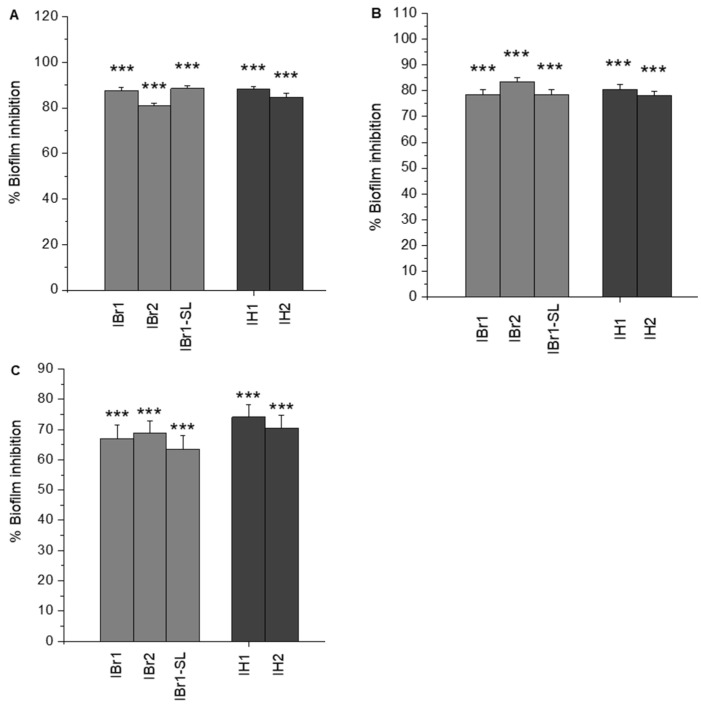
Evaluation of the inhibitory efficacy of plant extracts on three clinical isolates. (**A**) *P. aeruginosa* 39016; (**B**) *P. aeruginosa* PAK; (**C**) *P. aeruginosa* Mi162. Statistically significant effects are indicated with asterisks *** *p* < 0.001.

**Figure 4 pharmaceuticals-18-01824-f004:**
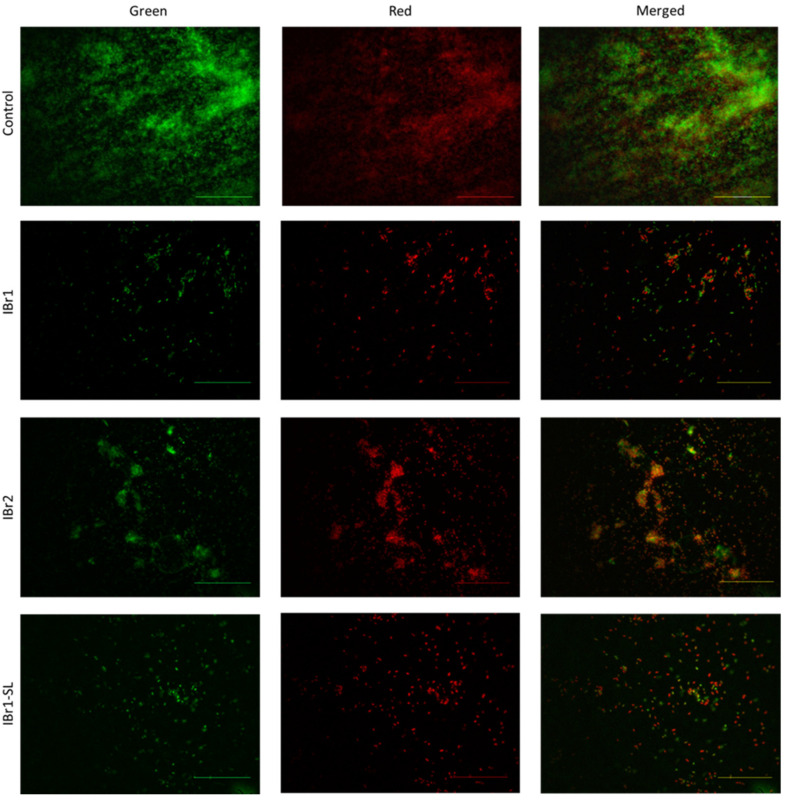
Evaluation of the viability of bacterial cells within the biofilms cultivated in the presence of *I. britannica* plant extracts. Bars = 50 μm.

**Figure 5 pharmaceuticals-18-01824-f005:**
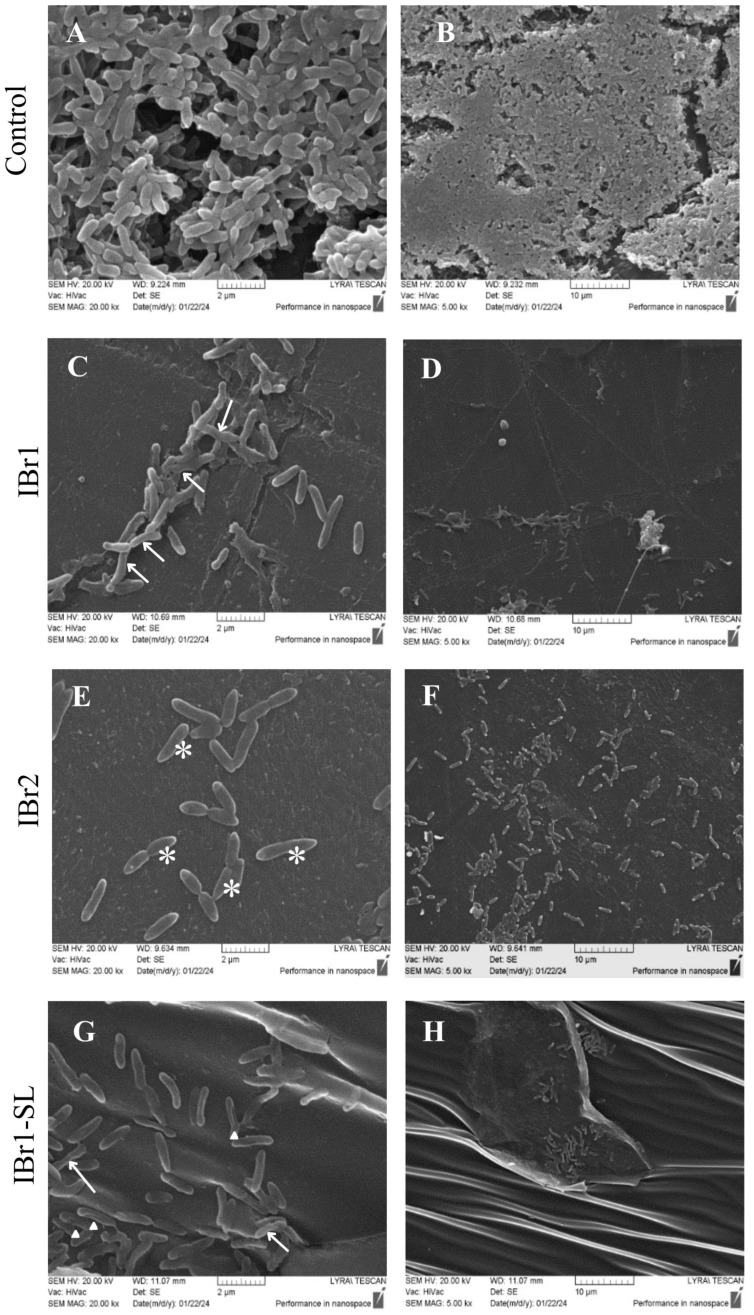
Evaluation of *P. aeruginosa* biofilms treated with plant extracts by scanning electron microscopy. (**A**,**B**) Control group; (**C**,**D**) biofilms cultivated with chloroform (**E**,**F**) methanol extracts and (**G**,**H**) lactone-enriched fraction from *I. britannica*. SEM magnifications: (**A**,**C**,**E**,**G**)—20,000; (**B**,**D**,**F**,**H**)—5000. Bars = 2–10 μm. White arrows point to invaginations, white stars—shape deformations, white triangle—longitudinal invaginations.

**Figure 6 pharmaceuticals-18-01824-f006:**
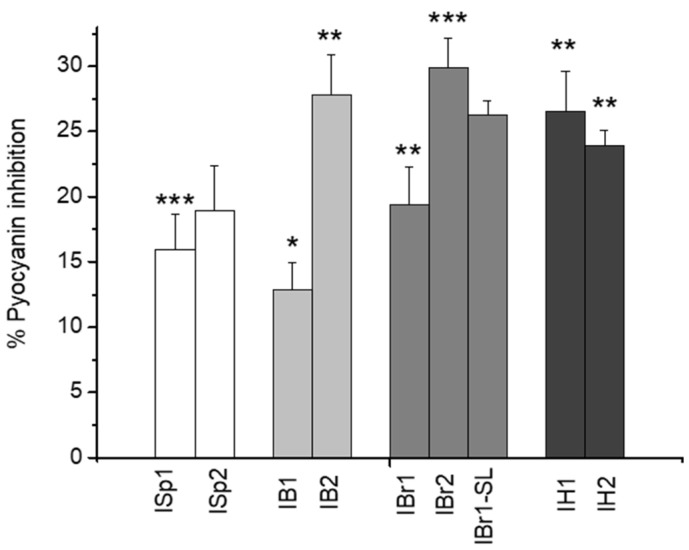
Evaluation of plant extracts on *P. aeruginosa* pyocyanin synthesis. Statistically significant effects are indicated with asterisks * *p* < 0.05, ** *p* < 0.01, *** *p* < 0.001.

**Figure 7 pharmaceuticals-18-01824-f007:**
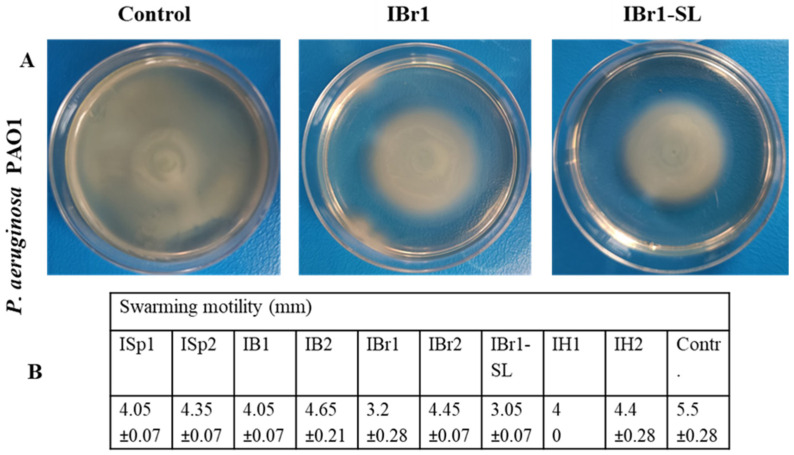
Effects of plant extracts on swarming motility. (**A**) Inhibition effects of plant extracts from *I. britannica* on swarming motility of *P. aeruginosa*. (**B**) Effects on swarming motility of all plants extracts.

**Figure 8 pharmaceuticals-18-01824-f008:**
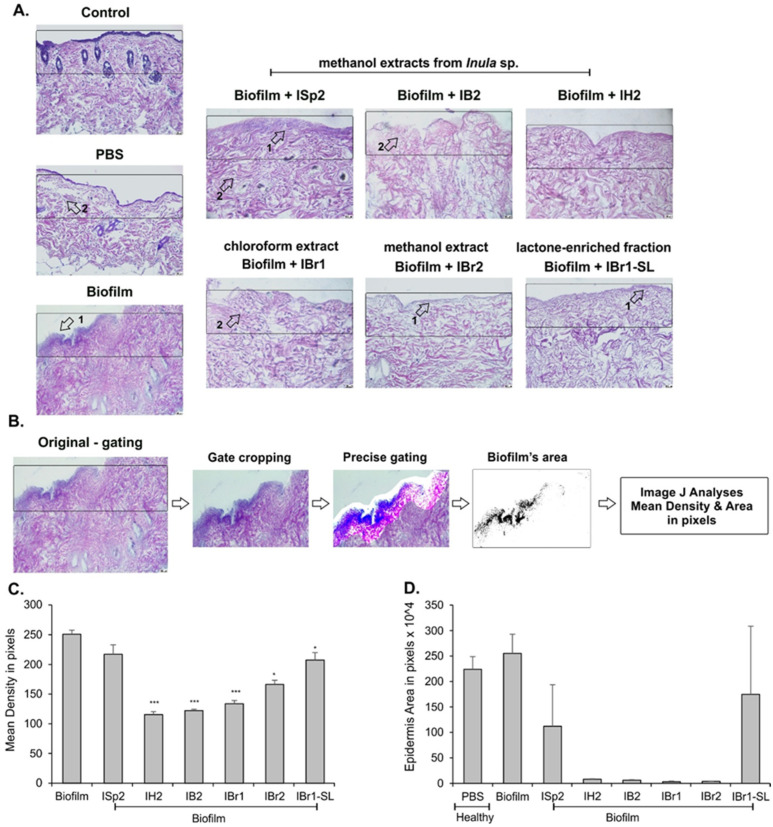
Effect of the Inula sp. extracts on *P. aeruginosa* biofilms formed at murine skin explant surface. H&E staining of murine skin explants collected after biofilm induction (for 24 h) and treatment with the extracts (100 µg/mL) (**A**). A control group shows the histology of health murine skin, and a PBS group indicates the skin explants cultured for 24 h and exposed to PBS without biofilm in vitro. The arrows in (**A**) show (1) biofilm formation and (2) loss of cellularity in the epidermis and dermis. The steps-based analyses of the H&E-stained murine skin explants by Image J software (version 1.8.0_172 (64 bit), National Institute of Health, USA) with precise gating shown blue-stained biofilm formation (**B**). The quantification of the mean density (**C**) and epidermal area (**D**) of skin explants. The data represent a mean ± SD of *n* = 6–9 samples per group. * *p* < 0.05, *** *p* < 0.001 when compared to the group with *P. aeruginosa* biofilm, ANOVA test.

**Figure 9 pharmaceuticals-18-01824-f009:**
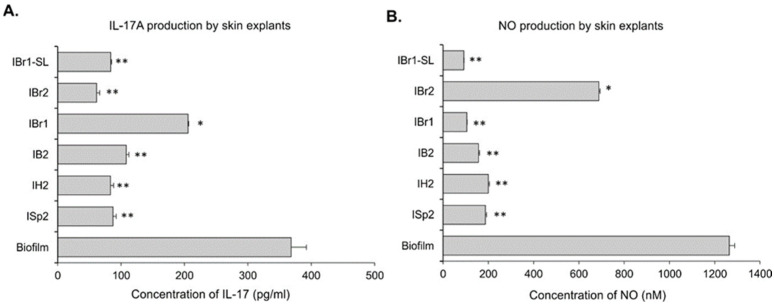
Inhibitory effect of the plant extracts on (**A**) the production of IL-17 and (**B**) NO by murine skin explants with *P. aeruginosa* biofilm. After the formation of the bacterial biofilm for 24 h, the murine skin explants were treated with 100 µg/mL plant extracts. The production and secretion in the supernatant of IL-17 were determined by ELISA and, for NO, were determined by colorimetric Griess reaction. The data represent a mean ± SD of *n* = 3 repeats per group; * *p* < 0.05, ** *p* < 0.01, vs. the biofilm group, ANOVA test.

**Table 1 pharmaceuticals-18-01824-t001:** Inhibition zones (mm) of plant extracts against *P. aeruginosa* extracellular protease production.

Protease Inhibition (mm) *								
ISp1	ISp2	IB1	IB2	IBr1	IBr2	IBr1-SL	IH1	IH2
13.5 ± 0.7	9.9 ± 0.14	9 ± 1.41	8 ± 1.41	9 ± 0.7	8.5 ± 1.41	9.75 ± 0.35	11.5 ± 2.12	8.5 ± 0.7

* Data represent (mean ± SD).

**Table 2 pharmaceuticals-18-01824-t002:** Strains used in the present study.

Strain	Characteristics	Reference
PAO1	A widely studied wound isolate from Melbourne, Australia.	[54]
PAK	*P. aeruginosa* clinical isolate, expresses pili, flagella, and glycosylation islands.	[83]
39016	*P. aeruginosa* clinical isolate, clone D by AT; serotype O11; carries distinctive pilA; subpopulation adapted to corneal infections; associated with severe infections; ST-235 agent of infections associated with the ocular cornea, resistant to clavulanic acid.	[84]
Mi162	*P. aeruginosa* clinical isolate, multidrug-resistant strain to 12 antibiotics, sensitive only to colistin, ciprofloxacin, and levofloxacin. Serotype 11.	[85]

## Data Availability

Data is contained within this article.

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
