# Peer review of "Interference of Pseudomonas aeruginosa Virulence Factors by Different Extracts from Inula Species"

_pharmaceuticals, 2025, doi:10.3390/ph18121824_

Round 1

Reviewer 1 Report

Comments and Suggestions for Authors

Dear Authors,

Congratulations on preparing such an interesting paper. Pseudomonas aeruginosa infections are among the most difficult clinical infections to treat. From my point of view, I am extremely pleased that research has been undertaken using plant extracts (from Inula sp. L.), which may prove helpful in inhibiting the growth of P. aeruginosa.

The work has been done carefully, guiding the reader from general information to details. I must admit that the inclusion of the supplementary section greatly improved my view of the research, especially with regard to the identification of the components of the plant extracts studied, which is essential for assessing the therapeutic potential of plant extracts.  Numerous figures in both the main part and the supplement (12 in total) facilitate familiarisation with the results of the research. The extensive discussion of the results shows that the authors planned and carried out the research procedures well and, with reference to the literature, correctly assessed and drew conclusions.

My minor comments relate to the methodology and preparation of extracts.

  • Do ‘aerial parts’ refer to flowering herb material, or are they exclusively herbaceous parts – leaves and stems?
  • Was extraction at room temperature assisted, e.g. by ultrasound, or was it a maceration process?
  • In line 534, chloroform appears in the methanol extract (which definitely should not be there).

I have no further comments on this work and congratulate the authors. I hope that this work will translate into further research into the potential of plant extracts and will be used to create preparations that reduce the possibility of infection with this pathogen.

Author Response

Dear reviewers,

we have carefully considered all critical remarks, comments and suggestions and have made all the necessary corrections/amendments accordingly. The corrections are outlined with track changes in the text. 

Reviewer 2 Report

Comments and Suggestions for Authors

1) Introduction:

i. A paragraph about the genus Inula should be included, summarizing the key previously reported information related to the genus and specifically the species tested in the current manuscript.

ii. The plant Family should be mentioned.

2) Results and Discussion:

i. The authors report the identification of several compounds (listed in Table S1) based solely on simple TLC comparison with authentic standards, GC/MS data, or, in one case, 1H NMR data. These techniques support only tentative identification and do not provide unambiguous confirmation of compound identity. The term “tentative” should be clearly stated throughout the discussion. Moreover, any available TLC plates, GC chromatograms, MS data, and 1H NMR spectra for these compounds should be included, at least, in the supplementary materials.

ii. The compounds mentioned in lines 134–143 are only tentatively identified, as no strong or supporting data are provided to justify these conclusions.

iii. Section 2.2. Cytotoxicity: A positive control should be included for comparison. Additionally, the ICâ‚…â‚€ values (in µg/mL) should be reported. The viability percentages presented in Figure 1 for the different extracts must also specify the concentrations at which they were measured. Please provide these data.

iv. Figure 1 legend: The fraction IBr1-SL should be clearly identified in the figure legend., and Figure quality should be improved.

v. Lines 189–192: This comparison is not valid. It does not confirm any meaningful conclusion, as chloroform extracts from different species naturally contain different compounds with varying activities.

vi. Line 203: The “enriched sesquiterpene lactone fraction” is mentioned, but no explanation is provided regarding how this fraction was prepared. Please include full details of the preparation method in the Experimental section. This information is also not provided in reference [12], so it must be described clearly in the manuscript.

vii. Also, mention the proposed or preivously reported mode of action of sesquiterpene lactones to enrich the discussion.

viii. All other tests (Biofilm inhibition, biofilm viability, pyocyanin inhibition, protease inhibition, biofilm on murine skin explants, IL-17 and NO production): No proper controls were included in these assays. It is essential to provide the results of both positive and negative controls to allow for meaningful and legitimate comparison. Without appropriate controls, the validity of these findings cannot be adequately assessed.

xi. Figur 2 legend: delete **P < 0.01 since it is not presented in the Figure.

x. Section 2.4: The first paragraph should be removed, as the information it contains has already been presented in the Introduction.

xi. Line 298: the difference between Figure 5A and 5B should be mentioned clearly ... i.e., the magnification power.

xii. Lines 353–357: It is inappropriate to draw such a conclusion solely by comparing these results with reported data on pure compounds, especially given that the compounds in your complex extracts are only partially and tentatively identified.

xiii. Arrange references in lines 480 and 484.

3. Materials and Methods:

i. Line 538 and Supplementary Materials section: Identifying compounds as 16-hydroxytriterpene fatty acid solely based on their Rf value (0.6–0.8) is not scientifically valid. More rigorous analytical data are required to confirm the compound’s identity.

ii. Line 524: Bulgarian population, should be corrected into flora!

iii. Supplementary Information: In the fractionation of the methanolic extracts, the volume of methanol used should be specified.

iv. Table 2: Describing some isolates as “resistant to clavulanic acid” may give the impression that clavulanic acid is an antibiotic, which is incorrect. Please revise the description to properly describe the isolates.

4. References:

i. Please double-check that all references are formatted correctly according to the journal’s guidelines.

Comments on the Quality of English Language

The manuscript contains numerous grammatical errors and unclear sentences. I recommend thorough revision by a native English speaker or professional language-editing service to enhance clarity and ensure the text is accessible to readers.

Author Response

(The authors gave the same response as above.)

Round 2

Reviewer 2 Report

Comments and Suggestions for Authors

Most of the comments have been addressed properly. However, the positive control in the cytotoxicity assay is still missing (not shown in the lines mentioned in the authors responses). This point was raised in my first round of revision but has not been added. 

Comments on the Quality of English Language

The manuscript contains numerous grammatical errors and unclear sentences. I recommend thorough revision by a native English speaker or professional language-editing service to enhance clarity and ensure the text is accessible to readers.